# DCWPSO: particle swarm optimization with dynamic inertia weight updating and enhanced learning strategies

Yibo Han[1], Meiting Lin[2], Ni Li[2,3], Qi Qi[1], Jinqing Li[1] and Qingxin Liu[1]

[1] School of Computer Science and Technology, Hainan University, Haikou, China
[2] School of Mathematics and Statistics, Hainan Normal University, Haikou, China
[3] Key Laboratory of Data Science and Smart Education of Ministry of Education, Hainan Normal University, Haikou, China



## ABSTRACT

Particle swarm optimization (PSO) stands as a prominent and robust meta-heuristic algorithm within swarm intelligence (SI). It originated in 1995 by simulating the foraging behavior of bird flocks. In recent years, numerous PSO variants have been proposed to address various optimization applications. However, the overall performance of these variants has not been deemed satisfactory. This article introduces a novel PSO variant, presenting three key contributions: First, a novel dynamic oscillation inertia weight is introduced to strike a balance between exploration and exploitation; Second, the utilization of cosine similarity and dynamic neighborhood strategy enhances both the quality of solution and the diversity of particle populations; Third, a unique worst-best example learning strategy is proposed to enhance the quality of the least favorable solution and consequently improving the overall population. The algorithm's validation is conducted using a test suite comprised of benchmarks from the CEC2014 and CEC2022 test suites on real-parameter single-objective optimization. The experimental results demonstrate the competitiveness of our algorithm against recently proposed state-of-the-art PSO variants and well-known algorithms.

## INTRODUCTION

Optimization algorithms are methodologies crafted to explore solutions for optimization problems, with the goal of identifying the most favorable solution based on a predefined criterion (*Engelbrecht, 2007*). The primary objective of the optimization process is to discover viable solutions that effectively address the given problem while satisfying any constraints. The intricate nature of specific optimization problems has contributed to the increasing prominence of meta-heuristic algorithms employed for solving optimization problems. Intelligent optimization algorithms have gained significant attention in recent years due to their ability to efficiently solve complex problems across various domains, such as, medicine, engineering, *etc.* These algorithms, inspired by natural phenomena or artificial intelligence principles, leverage advanced computational techniques to explore solution spaces and find optimal or near-optimal solutions. The intersection of

Corresponding authors
Ni Li, lini@hainnu.edu.cn
Qingxin Liu, qxliu@hainanu.edu.cn

nature-inspired computing, machine learning, and optimization has paved the way for the development of highly adaptive and efficient algorithms capable of addressing real-world challenges. One prominent category within this realm is metaheuristic algorithms, which encompass a diverse set of optimization techniques. Metaheuristics, such as genetic algorithm (GA) (*Wang, 2003*), artificial bee colony (ABC) (*Karaboga, 2010*), differential evolution (DE) (*Price, 1996*), simulated annealing (SA) (*Bertsimas & Tsitsiklis, 1993*), ant colony optimization (ACO) (*Dorigo, Birattari & Stutzle, 2006*) and Particle swarm optimization (PSO) (*Kennedy & Eberhart, 1995*), mimic natural processes or societal behaviors to iteratively improve candidate solutions.

PSO has emerged as a powerful optimization algorithm, drawing inspiration from the collective behavior of birds and fish. In 1995, Kennedy and Eberhart introduced the PSO algorithm, which navigates the problem space through the continuous adjustment of particles' velocity and position (*Kennedy & Eberhart, 1995*). Its simplicity, ease of implementation, and ability to explore high-dimensional solution spaces have made it a popular choice for solving complex optimization problems in various domains. Nevertheless, the investigation uncovered shortcomings in the PSO algorithm, particularly in terms of premature convergence and diminished convergence performance, especially as the optimization problem dimension increases (*Liang et al., 2006*; *Mendes, Kennedy & Neves, 2004*; *Qu, Suganthan & Das, 2012*).

The design of a rational and efficient evolutionary strategy has been a prevalent focus among researchers in the current year. Employing a single learning strategy may constrain the intelligence level of each particle, thereby diminishing the performance of PSO in addressing optimization problems with intricate fitness scenarios. Consequently, employing a hybrid learning strategy throughout the entire search process is considered to enhance the diversity of particle populations. In this study, we propose a dynamic oscillation inertia weight, cosine similarity based on dynamic neighborhood strategy and a worst-best example learning strategy based on PSO (DCWPSO), which introduces enhancements not only in the selection of inertia weights but also in the learning strategy. The contributions of this article have the following aspects:

- A novel dynamic oscillation inertia weight is proposed to strike a more effective balance between exploration and exploitation in the algorithm.
- A dynamic neighborhood strategy is proposed, deviating from the singular selection of *Pbest* and *Gbest*. Instead, particles are randomly chosen from their respective neighborhoods. This modification shows beneficial in enhancing both the diversity of particle motions and the diversity of particle populations. Additionally, the evolution of particles is fine-tuned by considering the cosine similarity between *Pbest* and *Gbest*.
- Worst-best example learning strategy is introduced to fine-tune the worst particle population, thereby enhancing the overall performance of the particle population.

The proposed algorithm undergoes analysis in terms of accuracy, stability, convergence and statistical analysis through experiments compared with PSO variants and well-known algorithms.

The remainder of this article is organized as follows: "Related Work" introduces classic PSO, parameter adjustment and strategy hybridization. "Proposed Algorithm" introduces the proposed algorithm. "Experimental Results and Analysis" discusses setup of experiments and analyzes experimental results. "Conclusions and Future Works" gives the conclusion and directions for future work.

## RELATED WORK

In this section, the primary focus is on the speed update mechanism of the canonical PSO algorithm and the key strategies employed by researchers to enhance the PSO algorithm.

### Canonical PSO

$$v_i(t+1) = \omega v_i(t) + c_1 r_1 (Pbest_i(t) - x_i(t)) + c_2 r_2 (Gbest(t) - x_i(t)) \qquad (1)$$
$$x_i(t+1) = x_i(t) + v_i(t+1) \qquad (2)$$

where $Pbest_i$ denotes the historical optimal solution of the particle $i$, $Gbest$ represents the historical optimal solution of the entire population, the position of the $i$th particle at the $t$th iteration is denoted as $x_i(t) = (x_{i1}(t), x_{i2}(t), \ldots, x_{iD}(t))$, the velocity of particle $i$ at the $t$th iteration is represented by $v_i(t) = (v_{i1}(t), v_{i2}(t), \ldots, v_{iD}(t))$. The parameter $\omega$, known as the inertial weight, regulates the impact of the previous velocity on the current velocity. Additionally, $r_1$ and $r_2$ are two randomly selected numbers from a uniform distribution [0,1]. $c_1$ represents the individual cognitive acceleration coefficient, while $c_2$ represents the social acceleration coefficient. These coefficients play a crucial role in shaping the behavior of the PSO algorithm.

### Parameter adjustment

Parameter adjustment in PSO primarily centers on the inertia weight coefficients $\omega$ and acceleration coefficients $c_1$, $c_2$.

The effectiveness of an optimization algorithm typically relies on achieving a balance between global search and local search across the entire search space. In light of this consideration, an inertia weight is introduced into the Eq. (1) for a particle. In previous studies, researchers have proposed various enhancements to inertia weights (*Chatterjee & Siarry, 2006*; *Arumugam & Rao, 2008*; *Al-Hassan, Fayek & Shaheen, 2006*; *Panigrahi, Pandi & Das, 2008*; *Feng et al., 2007*). In PSO, $c_1$ and $c_2$ are referred to as the cognitive component and the social component, respectively. They serve as stochastic acceleration coefficients responsible for adjusting the particle velocity with respect to $Pbest$ and $Gbest$. Hence, these two components play a crucial role in achieving the optimal solution rapidly and accurately. Some researchers have dedicated efforts to the selection of these two parameters (*Chen et al., 2018*; *Tian, Zhao & Shi, 2019*; *Kassoul, Belhaouari & Cheikhrouhou, 2021*; *Moazen et al., 2023*; *Sedighizadeh et al., 2021*; *Harrison, Engelbrecht & Ombuki-Berman, 2018*).

### Strategy hybridization

In general, there are two main ways to improve PSO through hybrid strategies, as shown in the following:

Improving PSO's performance by combining it with other search approaches. *Engelbrecht (2016)* introduced two adaptations of a parent-centric crossover PSO algorithm, leading to enhancements in solution accuracy compared to the original parent-centric PSO algorithms. The amalgamation of GA and PSO involves the partial integration of gene operations from GA, encompassing selection, crossover, and mutation, into PSO to enhance population diversity (*Molaei et al., 2021*; *Shi, Gong & Zhai, 2022*). Inspired by the bee-foraging search mechanism of the artificial bee colony algorithm, *Chen, Tianfield & Du (2021)* proposed a novel bee-foraging learning PSO (BFL-PSO) algorithm. *Singh, Singh & Houssein (2022)* proposed a novel hybrid approach known as the hybrid salp swarm algorithm with PSO (HSSA-PSO) for the exploration of high-quality optimal solutions in standard and engineering functions. *Hu, Cui & Bai (2017)* modified the constant acceleration coefficients by employing the exponential function, based on the combination of gravitational search algorithm (GSA) and PSO(PSO-GSA). *Khan & Ling (2021)* proposed a novel hybrid gravitational search PSO algorithm (HGSPSO). The fundamental idea behind this approach is to integrate the local search ability of GSA with the social thinking capability (Gbest) of PSO.

Incorporating topology in the PSO algorithm. *Liu & Nishi (2022)* proposed a novel strategy for exploring the neighbors of elite solutions. Additionally, the proposed algorithm was equipped with a constraint handling method to enable it to address constrained optimization problems. *Lee, Baek & Kim (2008)* proposed the repulsive PSO (RPSO) algorithm as a relatively recent heuristic search method. This algorithm was proposed as an effective approach to enhance the search efficiency for unknown radiative parameters. *Mousavirad & Rahnamayan (2020)* proposed a center-based velocity, incorporating a new component known as the "opening center of gravity factor", into the velocity update rule to formulate the center-based PSO (CenPSO). The center of gravity factor leveraged the center-based sampling strategy, a novel direction in population-based metaheuristics, particularly effective for addressing large-scale optimization problems. *Xu et al. (2019)* proposed the Two-Swarm Learning PSO (TSLPSO) algorithm, which was based on different learning strategies. One subpopulation constructed learning exemplars using the Dynamic Learning Strategy (DLS) to guide the local search of the particles, while the other subpopulation constructed learning exemplars using a comprehensive learning strategy to guide the global search. *Meng et al. (2022)* proposed a sorted particle swarm with hybrid paradigms to enhance optimization performance.

## PROPOSED ALGORITHM

The pseudo-code for the DCWPSO algorithm is presented in detail as Algorithm 1. The novelty of t proposed algorithm is encapsulated in the following discoveries: (1) A new dynamic oscillation inertia weight that better balance between global and local exploration. (2) The change involves altering the single method of selecting *Pbest* and *Gbest*, fostering increased population diversity. Additionally, cosine similarity is employed to assess the similarity between *Pbest* and *Gbest*, directing populations with low similarity to advance. (3) Strengthening the $p$ worst particles within the population to enhance the overall performance of the particle swarm.

**Algorithm 1  DCWPSO algorithm.**

**Input:** $FEs = 0, t = 1, MaxFEs, K, N, D, p, c1, c2, r1, r2$;

**Output:** $Gbest$;

Randomly initialize position vector $x_i (1 \leq i \leq N)$, velocity vector $v_i (1 \leq i \leq N)$;

1:    **while** $FEs < MaxFEs$ **do**

2:        Generate $\omega(t)$ by Eq. (3);

3:        Evaluate $x_i(t)(1 \leq i \leq N)$; $FEs = FEs + N$;

4:        Update $Pbest_i(t)(1 \leq i \leq N)$ and $Gbest(t)$;

5:        Sort $Pbest_i(t)(1 \leq i \leq N)$ according to the fitness values;

6:        **for** $i = 1$ to $N$ **do**

7:            **if** $FEs \geq 0.8*MaxFEs$ and $i$ is the worst $p$ agents index **then**

8:                Update the $i$th particle's velocity by Eq. (8);

9:            **else**

10:                Calculate the cosine similarity of $Pbest_i(t)$ and $Gbest$ neighborhoods, $cos\theta$;

11:                **if** $cos\theta < 0.5$ **then**

12:                    Update the $i$th particle's velocity by Algorithm 2;

13:                **else**

14:                    Update the ith particle's velocity by Eq. (1);

15:                **end if**

16:            **end if**

17:            Update $x_i(1 \leq i \leq N)$ based on Eq. (2);

18:        **end for**

19:        $t = t + 1$;

20:    **end while**

## Dynamic oscillation inertia weight

Within the context of PSO, the inertia weight holds significance as a pivotal parameter governing the dynamics of particle movement. Primarily, the role of the inertia weight lies in harmonizing the historical velocities of particles with the influences arising from individual experiences and group synergies. Traditional PSO employs a fixed-value inertia weight, limiting particles' ability to adapt to diverse environments and making the algorithm susceptible to local optima (*Kennedy & Eberhart, 1995*). Recognizing this limitation, *Shi & Eberhart (1999)* observed substantial enhancements in PSO performance by introducing a linearly changing inertia weight. While some investigations have utilized linear adaptive weights (*Xu & Pi, 2020*; *Van Den Bergh, 2001*; *Eberhart & Shi, 2000*), it has been acknowledged that, especially in the case of intricate optimization problems, nonlinear adaptive weights offer a better fitness to the environment and possess superior dynamic adjustment capabilities (*Ratnaweera, Halgamuge & Watson, 2004*; *Liu, Zhang & Tu, 2020*; *Chatterjee & Siarry, 2006*). This article introduces a novel nonlinear inertia weight represented by Eq. (3).

$$\omega(t) = r * \left( \frac{MaxFEs - FEs}{MaxFEs} \right)^2 * (\omega_{max} - \omega_{min}) + \omega_{min} \tag{3}$$

where, $r$ is a random number uniformly distributed in the interval of [0,1]. The parameters $\omega_{max}$ and $\omega_{min}$ are defined as 0.9 and 0.4, respectively. Function evaluations (FEs) denote the current number of evaluations, while maximum number of function evaluations (MaxFEs) represents the predefined maximum number of evaluations. Figure 1 depicts the trends of proposed and original weights curves.

From Fig. 1, it can be observed that with the progression of population iterations, the right side of Fig. 1 exhibits a linear decrease, while the left side of Fig. 1 demonstrates a fluctuating descent pattern. Incorporating this fluctuation strategy into the inertia weights aids the population in transitioning more frequently between the searching, following and scaping stage. This approach enhances the diversity of particle movement and contributes to an increased population diversity. This method of dynamic oscillation achieves a more optimal balance between the global and local search capabilities of particles, preventing them from becoming ensnared in local optima.

## Cosine similarity and dynamic neighborhood strategy

Throughout the evolution of the particle swarm algorithm, particles guide the population in the pursuit of the optimal solution by assimilating knowledge from historical personal best experiences (*Pbest*) and global best experiences (*Gbest*), however, depending solely on these two learning paradigms might not be adequate to convey the population with a comprehensive search knowledge. As the iteration progresses into its later stages, the *Pbest* and *Gbest* particles gradually converge towards the identified optimal regions, the particle population may incline towards local optimal solutions due to a shortage of search information. Hence, specific measures can be employed to assess the similarity between particles, followed by the selection of an appropriate learning paradigm. This ensures that all particles gain access to informative search information throughout the evolutionary process. There are two primary methods for assessing the similarity between two vectors in a high-dimensional space (*Qian et al., 2004*). Generally, whereas cosine similarity characterizes the relative distinction in direction, Euclidean distance characterizes the absolute distinction in objective value. In PSO algorithm, *Pbest* and *Gbest* mainly guide the movement of the particle swarm in the direction. Therefore, in this article we use the cosine similarity to compute the similarity between these two vectors to guide population evolution through angular information. Cosine similarity is independent of vector length, relying solely on the direction in which the vector is oriented. The mathematical expression for cosine similarity is denoted by Eq. (4). Figure 2 shows the cosine similarity of two particles from each of neighborhood.

$$cos(\theta) = \frac{M \cdot N}{\|M\|_2 \cdot \|N\|_2} \tag{4}$$

where $M = [y_1, y_2, \ldots, y_D]$ and $N = [z_1, z_2, \ldots, z_D]$. $M \cdot N$ denotes the inner product of vector $M$ and vector $N$. $\|M\|_2$ and $\|N\|_2$ represents the 2-Norm of vector $M$ and vector $N$.

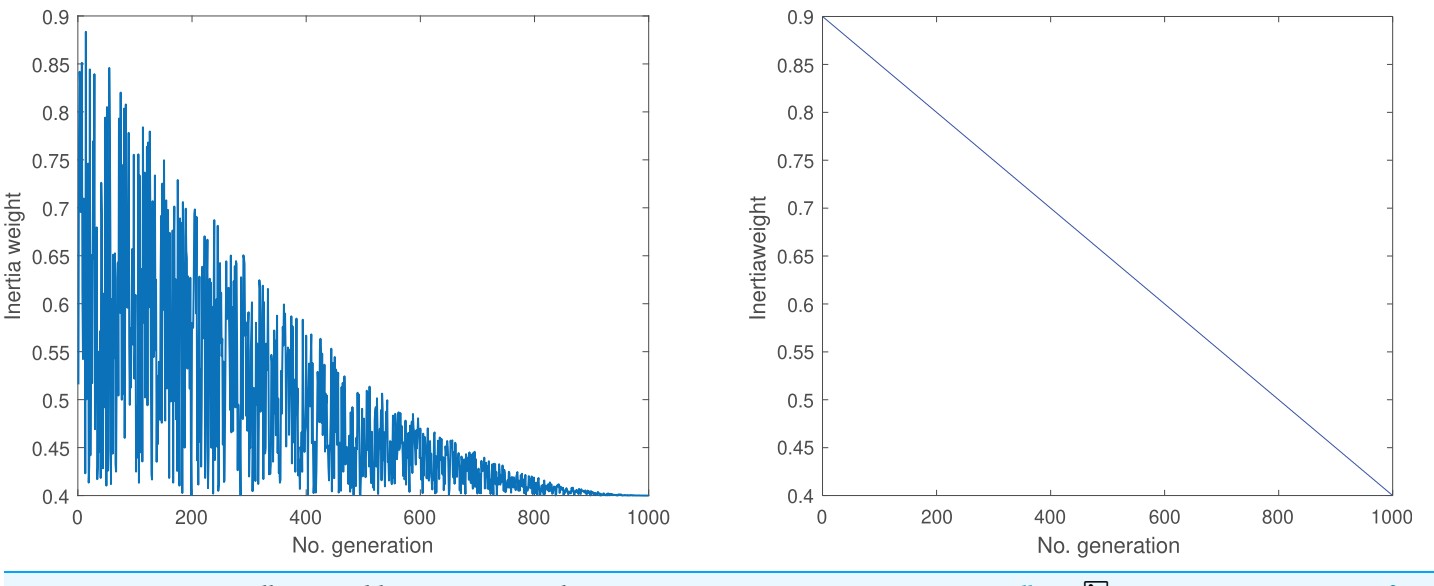

**Figure 1 Dynamic oscillation and linear inertia weight.**               

The characteristic of neighborhood has been employed in the variable neighborhood search (VNS) (*Mladenović & Hansen, 1997*) algorithm. It has the capability to discover the optimal solution within the current neighborhood and has the flexibility to escape the current neighborhood in search of a superior solution. In the classical PSO, the update of each particle is solely determined by the *Pbest* of an individual particle and the *Gbest* acquired from the entire particle swarm. This single selection method elevates the probability of particles being trapped in local optima. In this method, the closest *K* particles are selected to form a neighborhood by calculating the Euclidean distance of all particles from *Pbest* and *Gbest*, the equations are presented in Eqs. (5) and (6). A single particle is randomly chosen as the updated reference to guide the entire population within the respective neighborhood of *Pbest* and *Gbest*. Figure 2 shows the neighborhood of $Pbest_i(t)$ and *Gbest*, along with the particles within these neighborhoods, where the red particle represents *Gbest* and green particle represents $Pbest_i(t)$. The particles depicted in white signify the particles within the solution space. The blue particle and the red particle are randomly selected from the neighborhoods of $Pbest_i(t)$ and *Gbest*, respectively.

$$d_{i,Pbest_i(t)} = \sqrt{\sum_{i=1}^{D} (x_i - Pbest_i(t))^2} \tag{5}$$

$$d_{i,Gbest} = \sqrt{\sum_{i=1}^{D} (x_i - Gbest)^2}. \tag{6}$$

In general, a higher degree of similarity between two learning paradigms implies that their motion directions are more aligned, the positional difference is smaller, and the

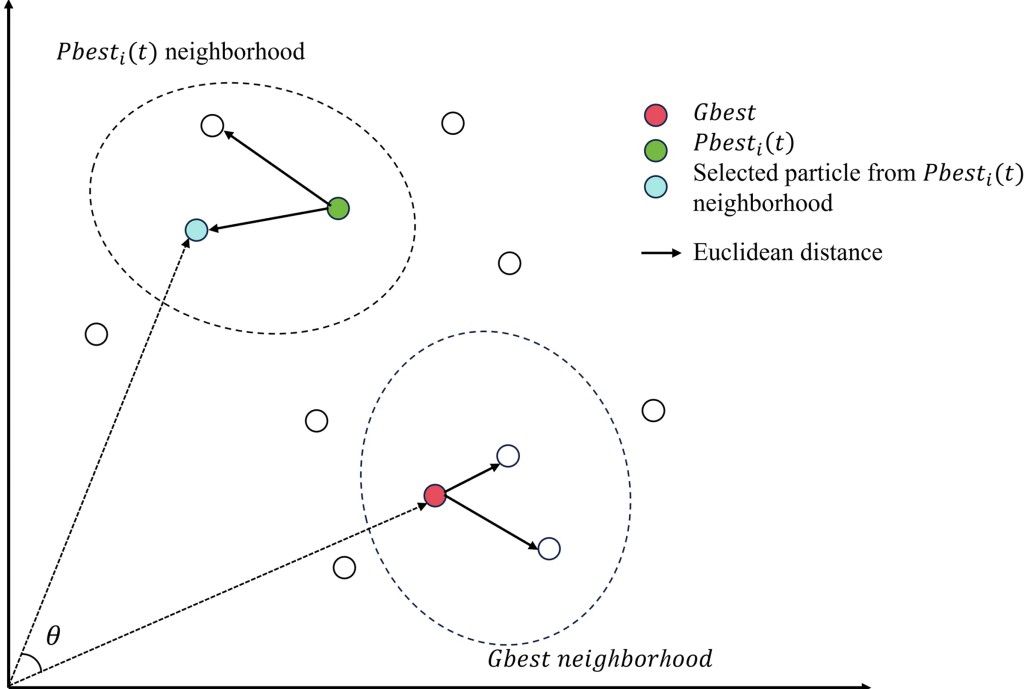

**Figure 2 Euclidean distance and cosine similarity in neighborhoods.**

number of feasible solutions contained between the paradigms is reduced. This results in less information that particles can learn from these paradigms, which hinders the evolutionary process. Conversely, a low degree of similarity indicates relatively independent motion directions and larger positional differences between the paradigms. In such cases, the paradigms encompass more feasible solutions, allowing particles to extract more valuable search information, facilitating the enhancement of solution quality. Therefore, in this method, the expandable range of *Pbest* and *Gbest* is augmented by elevating the particles' knowledge acquisition capability through the utilization of the neighborhood method when the similarity between Pbest and Gbest is high ($cos\theta < 0.5$). The equation for velocity update in cosine similarity dynamic neighborhood strategy is presented as Eq. (7).

$$v_i(t+1) = \begin{cases} \omega v_i(t) + c_1 r_1 (Pbest_i^l(t) - x_i(t)) + c_2 r_2 (Gbest^l(t) - x_i(t)) & cos\theta < 0.5 \\ Eq. \ (1) & cos\theta \geq 0.5 \end{cases} \quad (7)$$

$Pbest_i^l(t)$ and $Gbest^l(t)$ are particles randomly selected from the $Pbest_i(t)$ neighborhood and $Gbest(t)$ neighborhood, respectively.

According to the introduction above mentioned, the pseudo-code of the update method can be detailed as in Algorithm 2.

---

**Algorithm 2   Cosine similarity and dynamic neighborhood strategy.**

1:   **for** $i = 1$ to $N$ **do**

2:        Calculate the Euclidean distance from particle to $Pbest_i(t)$ and $Gbest$;

3:        Select the $K$ particles closest to $Pbest_i(t)$ to form a neighborhood;

4:   **end for**

5:   Select the $K$ particles closest to Gbest to form a neighborhood;

6:   Randomly select $Pbest_i^l(t)$ and $Gbest^l(t)$ from each of neighborhood respectively;

7:   Calculate the $cos\theta$ of $Pbest_i^l(t)$ and $Gbest^l(t)$ by Eq. (4);

8:   **if** $cos\theta < 0.5$ **then**

9:        Update the $i$th particle's velocity by Eq. (7);

10: **end if**

---

## Worst-best example learning strategy

In endeavors to enhance the collective performance of a group, the emphasis is often placed on elevating the capabilities of the least proficient individuals rather than solely promoting the top performers. This approach aims to catalyze substantial improvements in the overall group performance. The phenomenon referred to as the "cask effect", also known as the "short board effect", which implies that the water-holding capacity of a wooden bucket is determined not by its longest board but by its shortest board. Enhancing the length of the shortest board and removing constraints created by this short board can augment the water storage capacity of the wooden bucket. Similarly, within PSO, the overall performance of the entire population can be enhanced by adjusting the movement direction of the worst $p$ particles. The equation of velocity update as follows:

$$v_j^w(t + 1) = \omega v_j^w(t) + c_2 r_2 (Gbest(t) - x_j^w(t)) \tag{8}$$

where $x_j^w$ is the position of the particle $j$ from the worst particle neighborhood in the current population.

From Eq. (8), the update direction of a worst particle is solely oriented towards the global optimal experience and remains unaffected by individual optimal experiences. This facilitates a rapid improvement of the particle. These aspects enable the worst-best example learning strategy to enhance the quality of the population and mitigate the risk of falling into local optima.

## Complexity analysis

Time complexity is a key indicator of an algorithm's efficiency. The time complexity of the canonical PSO algorithm is $O(N \cdot D)$, where D is the dimension. The time complexity calculation of DCWPSO mainly includes two parts: dynamic neighborhood strategy and velocity and position update. For the dynamic neighborhood strategy, For the dynamic neighborhood strategy, firstly, the distances from each particle's position to $Pbest$ and

 

*Gbest* are calculated. Subsequently, the particles are sorted based on these distances, and the $k$ closest particles are selected. So the time complexity is $O(N^2)$. The time complexity of the update operation is consistent with that of canonical PSOs. In summary, the time complexity of DCWPSO is $O(N^2 + N \cdot D)$, which is slightly higher than that of $O(N \cdot D)$ of canonical PSO.

# EXPERIMENTAL RESULTS AND ANALYSIS

In this section, the proposed algorithm was rigorously compared with various PSO variants and other well-known algorithms on the CEC2014 and CEC2022 test suites, respectively. Comprehensive statistical analyses were performed to meticulously evaluate and elucidate the comparative performance of these algorithms.

## Setup of experiments

To validate DCWPSO, numerous experiments were conducted on complex functions extracted from the CEC2014 (*Liang, Qu & Suganthan, 2013*) and CEC2022 (*Yazdani et al., 2021*) test suites. This choice was made due to the high complexity of functions within the CEC2014 test suites compared to classical functions, rendering them notably challenging to solve. In CEC2014, 30 functions can be divided into four types relying on their properties, that is, unimodal functions ($f_1 \sim f_3$), simple multimodal functions ($f_4 \sim f_{16}$), hybrid functions ($f_{17} \sim f_{22}$) and composition functions ($f_{23} \sim f_{30}$). Additionally, 12 functions from the latest CEC2022 suites were selected to further assess the algorithm's capability in addressing contemporary complex optimization problems. Specifically, $f_1$ is a unimodal function, $f_2 \sim f_5$ are basic functions, $f_6 \sim f_8$ are hybrid functions, and $f_9 \sim f_{12}$ are composition functions.

In this experiment, the proposed algorithm is compared against several classical and advanced PSO variants, as well as other well-known algorithms. The PSO variants include APSO (*Zhan et al., 2009*) PSO-DLS (*Ye, Feng & Fan, 2017*), XPSO (*Xia et al., 2020*), PSO-CL (*Liang, Zhao & Li, 2021*), and ADFPSO (*Yu, Tong & Xia, 2022*). The well-known algorithms include PSO (*Kennedy & Eberhart, 1995*), ICSPM2 (*Abed-alguni & Paul, 2022*), DGWO (*Abed-alguni & Barhoush, 2018*), AGWO (*Ma et al., 2024*), and PO (*Lian et al., 2024*). Notably, the AGWO and PO algorithms have recently demonstrated strong performance on the CEC2022 test suites. The details of these algorithms are presented in Tables 1 and 2.

Each function is independently run 30 times, To accurately reproduce the performance of the comparison algorithms, the termination criteria are defined as follows: for CEC2014, the maximum number of evaluations is set to $D \times 10^4$, where $D$ is the dimension. For CEC2022, the termination criterion is set to the maximum number of iterations, which is set to $10^4$. Search range is $[-100, 100]^D$.

## Comparisons of the solution accuracy and stability

The proposed algorithm and comparison algorithms are tested on $30 - D$ CEC2014 test suites and $20 - D$ CEC2022 test suites. Tables 3 and 4 list the mean and standard deviation value for each function and the best results are denoted in bold.

**Table 1 The parameter settings of PSO variants.**

| Algorithm | Years | Parameter setting |
|---|---|---|
| APSO (*Zhan et al., 2009*) | 2009 | $N = 30, \omega = 0.9, c_1 = c_2 = 2, \delta = [0.05, 0.1]$ |
| PSO-DLS (*Ye, Feng & Fan, 2017*) | 2017 | $N = 30, \omega = [0.4, 0.9], c_1 = c_2 = 1.49445$ |
| XPSO (*Xia et al., 2020*) | 2020 | $N = 30, \omega = [0.4, 0.9], \eta = 0.2, Stag_{max} = 5, p = 0.2$ |
| PSO-CL (*Liang, Zhao & Li, 2021*) | 2021 | $N = 30, \omega = [0.4, 0.9], c_1 = c_2 = 2$ |
| ADFPSO (*Yu, Tong & Xia, 2022*) | 2022 | $N = 30, \omega = [0.4, 0.9], c_1$ and $c_2$ are adjustable, $K = 2$ |
| DCWPSO | – | $N = 30, \omega = [0.4, 0.9], c_1$ and $c_2 = 2, K = 2, p = 2$ |

**Table 2 The parameter settings of well-known algorithms.**

| Algorithm | Years | Parameter setting |
|---|---|---|
| PSO (*Kennedy & Eberhart, 2004*) | 2004 | $N = 30, \omega = 0.9, c_1 = c_2 = 2$ |
| ICSPM2 (*Abed-alguni & Paul, 2022*) | 2022 | $N = 30, s = 12, M_f = 100, M_r = 0.2$ |
| DGWO (*Abed-alguni & Barhoush, 2018*) | 2018 | $N = 30, s = 10, M_f = 50, M_r = 0.2$ |
| AGWO (*Ma et al., 2024*) | 2024 | $N = 30$ |
| PO (*Lian et al., 2024*) | 2024 | $N = 30$ |
| DCWPSO | – | $N = 30, \omega = [0.4, 0.9], c_1$ and $c_2 = 2, K = 2, p = 2$ |

**Table 3 Comparison results of DCWPSO with other PSO variants on CEC2014 test set ($D = 30$). The best values are highlighted in bold.**

| Function | Metrics | APSO | PSO-DLS | XPSO | PSO-CL | ADFPSO | DCWPSO |
|---|---|---|---|---|---|---|---|
| $f_1$ | Mean | 2.10743E+06 | 4.19287E+06 | 4.09250E+06 | 5.99798E+07 | 3.01042E+06 | **5.25727E+05** |
|  | Std | 2.26275E+06 | 4.01228E+06 | 4.92657E+06 | 2.23995E+07 | 1.46053E+06 | **3.80136E+05** |
| $f_2$ | Mean | 1.40231E+04 | 6.24251E+03 | 3.74607E+03 | 8.58606E+07 | 9.68756E+03 | **2.00156E+02** |
|  | Std | 1.37856E+04 | 8.12712E+03 | 2.83005E+03 | 2.03640E+08 | 7.21276E+03 | **3.12379E−01** |
| $f_3$ | Mean | 1.24303E+04 | 1.57451E+03 | 6.69169E+02 | 3.10723E+04 | 8.50665E+02 | **3.01152E+02** |
|  | Std | 1.19470E+04 | 1.29977E+03 | 3.73894E+02 | 7.93087E+03 | 3.25013E+02 | **1.78408E+00** |
| $f_4$ | Mean | 4.77208E+02 | 6.00856E+02 | 5.26197E+02 | 6.72202E+02 | 5.17765E+02 | **4.64193E+02** |
|  | Std | 4.86314E+01 | 4.47689E+01 | 4.00371E+01 | 4.46627E+01 | **1.04599E+01** | 4.38210E+01 |
| $f_5$ | Mean | **5.20032E+02** | 5.20539E+02 | 5.20953E+02 | 5.20944E+02 | 5.20938E+02 | 5.20379E+02 |
|  | Std | **2.55358E−02** | 1.56973E−01 | 8.62988E−02 | 5.50932E−02 | 4.92628E−02 | 3.22722E−01 |
| $f_6$ | Mean | 6.17837E+02 | 6.16320E+02 | 6.07751E+02 | 6.21067E+02 | **6.00156E+02** | 6.14984E+02 |
|  | Std | 3.17488E+00 | 2.69610E+00 | 3.43317E+00 | 2.30242E+00 | **4.43376E-01** | 4.10208E+00 |
| $f_7$ | Mean | 7.00051E+02 | 7.01681E+02 | 7.00016E+02 | 7.07414E+02 | 7.00001E+02 | **7.00000E+02** |
|  | Std | 4.66498E−02 | 3.52745E−01 | 1.98110E−02 | 1.74946E+00 | 2.79975E−03 | **1.80002E−03** |
| $f_8$ | Mean | **8.00000E+02** | 8.87573E+02 | 8.35686E+02 | 9.99897E+02 | 8.16070E+02 | 9.08516E+02 |
|  | Std | **5.22329E−06** | 2.06257E+01 | 1.08796E+01 | 1.33073E+01 | 4.54755E+00 | 1.90488E+01 |
| $f_9$ | Mean | 9.94373E+02 | 1.00560E+03 | 9.52069E+02 | 1.11240E+03 | **9.18517E+02** | 1.03604E+03 |
|  | Std | 2.33145E+01 | 2.22062E+01 | 1.44427E+01 | 1.43172E+01 | **6.92760E+00** | 3.15291E+01 |
| $f_{10}$ | Mean | **1.00000E+03** | 5.11840E+03 | 1.82808E+03 | 7.33732E+03 | 1.86736E+03 | 2.74601E+03 |
|  | Std | **1.29506E−02** | 8.23218E+02 | 3.66027E+02 | 3.97352E+02 | 4.01896E+02 | 4.79595E+02 |

(Continued)

| Function | Metrics | APSO | PSO-DLS | XPSO | PSO-CL | ADFPSO | DCWPSO |
|---|---|---|---|---|---|---|---|
| $f_{11}$ | Mean | 3.90476E+03 | 6.06016E+03 | 3.59387E+03 | 8.01932E+03 | **2.72688E+03** | 4.28585E+03 |
|  | Std | 5.29050E+02 | 8.91069E+02 | 5.83328E+02 | **2.87878E+02** | 5.70080E+02 | 7.07878E+02 |
| $f_{12}$ | Mean | 1.20251E+03 | 1.20114E+03 | 1.20174E+03 | 1.20246E+03 | 1.20247E+03 | **1.20019E+03** |
|  | Std | 2.73418E−01 | 3.51232E−01 | 9.32763E−01 | 2.32045E-01 | 3.42228E−01 | **8.54987E−02** |
| $f_{13}$ | Mean | **1.30016E+03** | 1.30031E+03 | 1.30029E+03 | 1.30065E+03 | 1.30016E+03 | 1.30041E+03 |
|  | Std | **3.67055E−02** | 9.19245E−02 | 7.98839E−02 | 1.15361E−01 | 4.34289E−02 | 9.83510E−02 |
| $f_{14}$ | Mean | 1.40026E+03 | 1.40027E+03 | 1.40035E+03 | 1.40065E+03 | 1.40025E+03 | **1.40022E+03** |
|  | Std | **3.35147E−02** | 9.45794E−02 | 1.92893E−01 | 2.18700E−01 | 1.01548E−01 | 7.08499E−02 |
| $f_{15}$ | Mean | 1.50467E+03 | 1.51282E+03 | 1.50569E+03 | 1.52885E+03 | **1.50410E+03** | 1.50525E+03 |
|  | Std | 2.64882E+00 | 2.55177E+00 | 2.31131E+00 | 3.70031E+00 | **1.07817E+00** | 1.92227E+00 |
| $f_{16}$ | Mean | **1.60979E+03** | 1.61175E+03 | 1.61077E+03 | 1.61282E+03 | 1.60981E+03 | 1.61138E+03 |
|  | Std | 1.04163E+00 | 6.19166E−01 | 9.09076E−01 | **2.01469E−01** | 1.21942E+00 | 7.54887E−01 |
| $f_{17}$ | Mean | 1.18497E+05 | 6.46229E+05 | 2.33900E+05 | 1.81444E+06 | 1.40970E+05 | **8.22408E+04** |
|  | Std | **5.30145E+04** | 5.47877E+05 | 2.40759E+05 | 7.10083E+05 | 9.39262E+04 | 9.14161E+04 |
| $f_{18}$ | Mean | 1.45735E+04 | 2.37362E+05 | **4.06654E+03** | 3.88846E+06 | 1.06793E+04 | 4.14324E+03 |
|  | Std | 4.42497E+04 | 6.26807E+05 | **2.37243E+03** | 1.91290E+06 | 3.76555E+04 | 3.68983E+03 |
| $f_{19}$ | Mean | 1.90562E+03 | 1.91606E+03 | 1.90797E+03 | 1.92116E+03 | **1.90559E+03** | 1.91261E+03 |
|  | Std | 9.09610E−01 | 1.09175E+01 | 1.66442E+00 | 2.75543E+00 | **1.26986E+00** | 1.45659E+01 |
| $f_{20}$ | Mean | 2.37369E+03 | 2.37553E+03 | 2.50543E+03 | 6.48692E+03 | 2.62925E+03 | **2.20314E+03** |
|  | Std | 2.20740E+02 | 1.07684E+02 | 2.38146E+02 | 3.26125E+03 | 6.07135E+02 | **7.36162E+01** |
| $f_{21}$ | Mean | 1.94832E+05 | 6.60844E+04 | 7.01337E+04 | **6.48692E+03** | 4.83188E+04 | 2.93569E+04 |
|  | Std | 2.94413E+05 | 5.98596E+04 | 1.41651E+05 | 2.89190E+05 | 2.97001E+04 | **1.90201E+04** |
| $f_{22}$ | Mean | 2.90899E+03 | 2.58324E+03 | 2.50481E+03 | 2.76956E+03 | **2.42111E+03** | 2.66276E+03 |
|  | Std | 2.06202E+02 | 1.42353E+02 | 1.24514E+02 | 1.36904E+02 | **9.05582E+01** | 2.32602E+02 |
| $f_{23}$ | Mean | 2.61528E+03 | 2.62159E+03 | 2.61573E+03 | 2.63155E+03 | 2.61524E+03 | **2.61524E+03** |
|  | Std | 3.37140E−02 | 5.46025E+00 | 1.96627E−01 | 3.18393E+00 | 8.58264E−04 | **1.77000E−12** |
| $f_{24}$ | Mean | 2.63522E+03 | 2.64368E+03 | 2.62443E+03 | 2.61319E+03 | **2.60417E+03** | 2.62128E+03 |
|  | Std | 5.06717E+00 | 6.03877E+00 | **1.62088E+00** | 1.38638E+01 | 6.67843E+00 | 4.39020E+00 |
| $f_{25}$ | Mean | 2.70990E+03 | 2.71114E+03 | 2.70945E+03 | 2.71046E+03 | **2.70433E+03** | 2.71047E+03 |
|  | Std | 4.65896E+00 | 2.79212E+00 | **2.03228E+00** | 7.74408E+00 | 4.23654E−01 | 2.87615E+00 |
| $f_{26}$ | Mean | 2.76314E+03 | 2.71035E+03 | 2.76014E+03 | **2.70066E+03** | 2.78010E+03 | 2.76021E+03 |
|  | Std | 8.57504E+01 | 3.06707E+01 | 4.96974E+01 | **9.32209E−02** | 4.06561E+01 | 4.96262E+01 |
| $f_{27}$ | Mean | 3.45960E+03 | 3.39347E+03 | 3.27641E+03 | 3.36547E+03 | **3.05698E+03** | 3.33054E+03 |
|  | Std | 2.30629E+02 | 1.86152E+02 | 1.37403E+02 | 1.46638E+02 | **5.35099E+01** | 2.44008E+02 |
| $f_{28}$ | Mean | 4.06853E+03 | 4.47938E+03 | 4.23487E+03 | 4.28153E+03 | **3.75112E+03** | 5.24137E+03 |
|  | Std | 4.09894E+02 | 6.40743E+02 | 5.52933E+02 | 3.49932E+02 | **1.50121E+02** | 7.86963E+02 |
| $f_{29}$ | Mean | 1.99976E+06 | 6.61187E+06 | 4.41827E+06 | 3.13958E+05 | 6.00518E+03 | **4.15505E+03** |
|  | Std | 3.67879E+06 | 1.36228E+07 | 1.01681E+07 | 1.80512E+05 | 3.40260E+03 | **4.40251E+02** |
| $f_{30}$ | Mean | 7.11373E+03 | 1.34335E+04 | 6.56060E+03 | 3.99887E+04 | 5.35929E+03 | **5.21838E+03** |
|  | Std | 1.86650E+03 | 9.16392E+03 | 1.07930E+03 | 1.75699E+04 | 8.60589E+02 | **7.86773E+02** |
| Count (Mean) |  | 5 | 0 | 1 | 2 | 10 | **12** |
| Count (Std) |  | 6 | 0 | 3 | 3 | 8 | **10** |

**Table 4 Comparison results of DCWPSO with well-known algorithms on CEC2022 test set ($D = 20$).** The best values are highlighted in bold.

| Function | Metrics | PSO | ICSPM2 | DGWO | AGWO | PO | DCWPSO |
|---|---|---|---|---|---|---|---|
| $f_1$ | Mean | 6.781E+03 | 6.408E+04 | 2.020E+04 | 1.822E+04 | 9.523E+02 | **3.000E+02** |
| | Std | 1.469E+03 | 9.943E+03 | 2.275E+03 | 3.593E+03 | 5.052E+02 | **0.000E+00** |
| $f_2$ | Mean | 5.870E+02 | 3.400E+03 | 9.057E+02 | 5.706E+02 | 4.606E+02 | **4.485E+02** |
| | Std | 3.417E+01 | 6.362E+02 | 7.511E+01 | 2.643E+01 | 2.439E+01 | **1.859E+01** |
| $f_3$ | Mean | 6.218E+02 | 6.918E+02 | 6.761E+02 | 6.286E+02 | 6.481E+02 | **6.143E+02** |
| | Std | **2.060E+00** | 8.371E+00 | 6.014E+00 | 2.788E+00 | 9.292E+00 | 1.463E+01 |
| $f_4$ | Mean | 9.202E+02 | 1.042E+03 | 9.497E+02 | 9.060E+02 | 8.845E+02 | **8.813E+02** |
| | Std | **6.771E+00** | 1.451E+01 | 1.184E+01 | 1.359E+01 | 1.803E+01 | 1.790E+01 |
| $f_5$ | Mean | 1.868E+03 | 6.107E+03 | 3.279E+03 | 1.409E+03 | 2.058E+03 | **1.272E+03** |
| | Std | 1.516E+02 | 2.588E+02 | 2.021E+02 | **9.784E+01** | 3.010E+02 | 6.110E+02 |
| $f_6$ | Mean | 6.016E+07 | 2.410E+09 | 5.062E+07 | 2.577E+07 | 5.947E+03 | **5.104E+03** |
| | Std | 1.858E+07 | 5.336E+08 | 1.462E+07 | 1.126E+07 | 5.042E+03 | **3.527E+03** |
| $f_7$ | Mean | 2.080E+03 | 2.270E+03 | 2.278E+03 | 2.096E+03 | 2.127E+03 | **2.079E+03** |
| | Std | **8.002E+00** | 4.920E+01 | 5.409E+01 | 2.321E+01 | 2.649E+01 | 3.810E+01 |
| $f_8$ | Mean | 2.245E+03 | 2.668E+03 | 2.418E+03 | 2.249E+03 | **2.240E+03** | 2.272E+03 |
| | Std | **5.291E+00** | 1.414E+02 | 8.020E+01 | 9.554E+00 | 9.899E+00 | 7.282E+01 |
| $f_9$ | Mean | 2.507E+03 | 3.126E+03 | 2.877E+03 | 2.539E+03 | 2.490E+03 | **2.481E+03** |
| | Std | 3.642E+00 | 1.058E+02 | 6.471E+01 | 2.144E+01 | 1.366E+01 | **3.045E−04** |
| $f_{10}$ | Mean | 4.549E+03 | 4.380E+03 | 4.801E+03 | 2.601E+03 | **2.541E+03** | 3.300E+03 |
| | Std | 1.590E+03 | 1.227E+03 | 1.756E+03 | 1.147E+02 | **8.099E+01** | 4.886E+02 |
| $f_{11}$ | Mean | 3.934E+03 | 9.131E+03 | 6.346E+03 | 3.926E+03 | 2.954E+03 | **2.910E+03** |
| | Std | 3.698E+02 | 8.764E+02 | 3.781E+02 | 3.020E+02 | 8.443E+01 | **7.000E+01** |
| $f_{12}$ | Mean | 3.008E+03 | **2.900E+03** | 3.836E+03 | 2.932E+03 | 2.983E+03 | 3.023E+03 |
| | Std | 1.341E+01 | **1.865E−09** | 1.453E+02 | 5.966E+00 | 2.927E+01 | 5.921E+01 |
| Count (Mean) | | 0 | 1 | 0 | 0 | 2 | **9** |
| Count (Std) | | 4 | 1 | 0 | 1 | 1 | **5** |

In Table 3, The best values are highlighted in bold. Among the 30 test functions, DCWPSO achieves the highest number of best-performing functions, with 12 in terms of mean and 10 in terms of standard deviation. Both of these counts are the highest among all the algorithms. The proposed algorithm achieves the top ranking across all unimodal functions ($f_1 \sim f_3$), underscoring its effectiveness in solving such functions. For the evaluation of 13 simple multimodal functions ($f_4 \sim f_{16}$), the proposed algorithm secures the third position, following APSO and ADFPSO. Notably, in addressing hybrid functions ($f_{17} \sim f_{22}$), the proposed algorithm attains the first rank in $f_{17}$, $f_{18}$, $f_{20}$ and $f_{21}$, outperforming other PSO variants. For composition functions ($f_{23} \sim f_{30}$), DCWPSO is ranked second among all algorithms, demonstrating notable proficiency in solving $f_{23}$, $f_{29}$ and $f_{30}$. In terms of solution accuracy across the 30 test functions, DCWPSO ranks first in 12 of them, securing the top overall rank with a significant advantage over other algorithms.

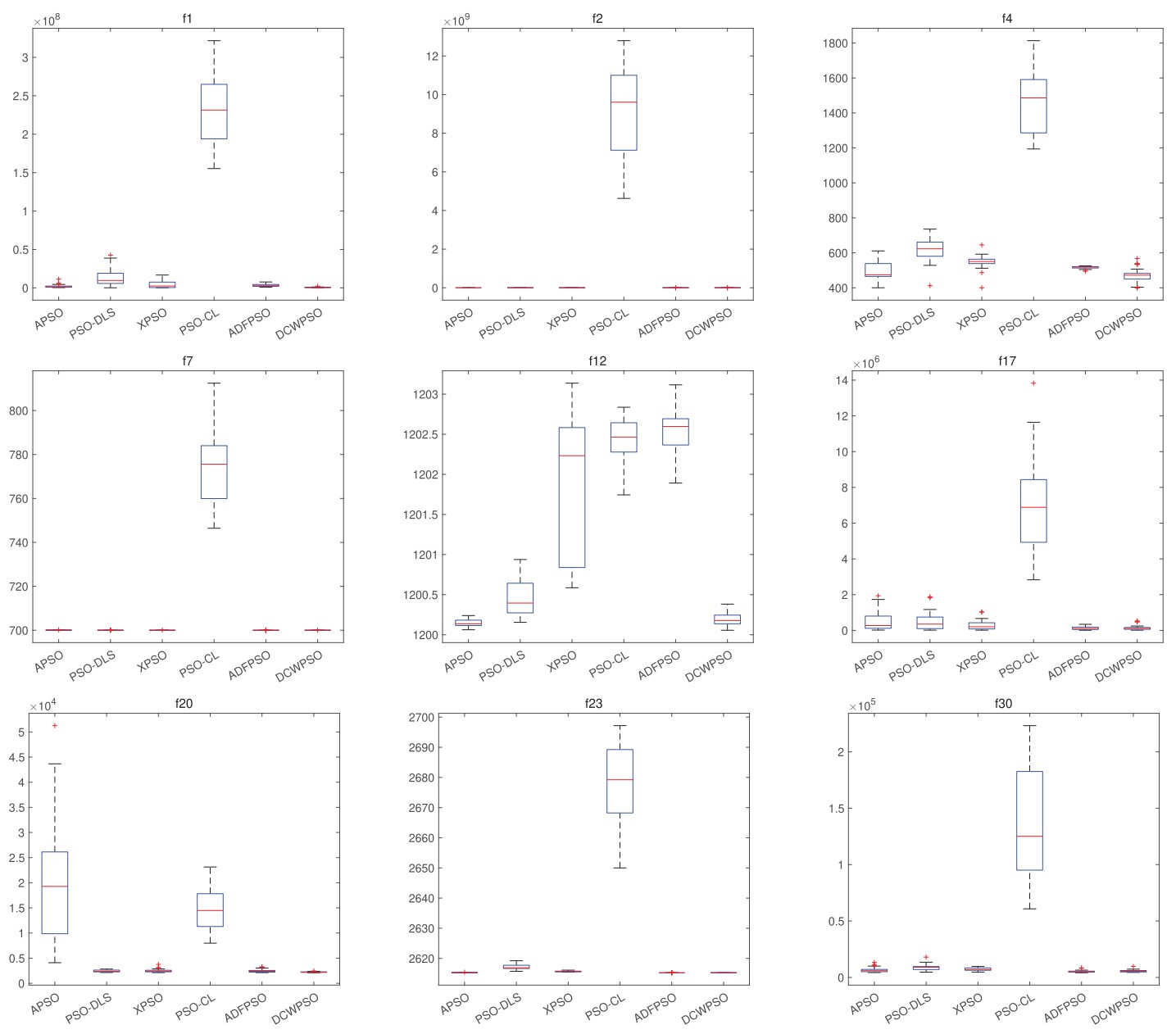

**Figure 3 Stability effect of DCWPSO compared with PSO variants in the CEC2014 $(30 - D)$.**

DCWPSO also demonstrates commendable stability in the comparisons. Table 3 presents the standard deviation of the outcomes from 30 independent executions for each of the 30 test functions in CEC2014. Figure 3 illustrates a subset of these test functions through box plots depicting their results. The proposed algorithm secured the first rank in 10 out of all the tested functions. It not only excels in accuracy but also demonstrates notably competitive stability, holding a substantial advantage in comprehensive performance when compared with other PSO variants.

As shown in Table 4, The best values are highlighted in bold. Among the 12 test functions, DCWPSO achieves the highest number of best-performing functions, with 9 in terms of mean and 5 in terms of standard deviation. Both of these counts are the highest among all the algorithms. The algorithm proposed in this article rank first in the unimodal function $f_1$ and exhibited the lowest standard deviation also. This finding indicates that the algorithm has significant advantages in solving unimodal functions. Similarly, for the basic functions ($f_2 \sim f_5$), the proposed algorithm consistently ranked first. This indicates that the proposed algorithm possesses a distinct advantage in solving basic functions. For the hybrid functions ($f_6 \sim f_8$), the proposed algorithm achieves a strong performance for function $f_6$ and $f_7$, but a mediocre performance for functions $f_8$. For the composition functions ($f_9 \sim f_{12}$), The proposed algorithm is ranked first in two out of four functions. This suggests that the DCWPSO algorithm has significant advantages in addressing certain challenging composition functions.

Among the 12 test functions of CEC2022, the proposed algorithm achieved the highest mean value rankings in nine test functions and the lowest standard deviation rankings in five test functions. This demonstrates that the proposed algorithm excels in both accuracy and stability.

## Comparisons of convergence performance

This experiment is conducted on the CEC2014 test suite to scrutinize the convergence performance of the DCWPSO algorithm across four types of functions. To accentuate the performance of the DCWPSO algorithm, only nine convergence curve figures featuring representative functions are selected. Specific experimental results are shown in the figures below.

Figure 4 displays the convergence curves of the proposed algorithm alongside those of the comparison algorithms for the CEC2014 functions. The consistent outperformance of $f_1$ and $f_2$ function over other PSO variants from the beginning to the end of the iteration suggests that the proposed algorithm has a clear advantage in solving unimodal functions of this type. For function $f_4$, the proposed algorithm consistently achieves superior solutions compared to other PSO variants. Analysis of the iteration curves shows its capability to avoid local optima in later stages and find global optimum solutions, highlighting its strength in escaping local optima. For function $f_7$, the proposed algorithm excels in rapidly achieving superior solutions. While most other PSO variants converge to nearly identical solution, DCWPSO ans APSO requires fewer iterations. For function $f_{12}$, In the initial stages, the proposed algorithm may not discover as optimal a solution as the APSO algorithm. However, as iterations progress, it demonstrates the ability to escape local optima and achieve superior solutions more rapidly than several other PSO variants, underscoring the robust tuning capability of DCWPSO. For function $f_{17}$ and $f_{20}$, the proposed algorithm consistently explores new globally optimal solutions during the initial and middle stages of iteration, with gradual convergence observed in later stages. This behavior indicates the algorithm's proficiency in effectively tackling hybrid functions as well. Similarly, for function $f_{23}$ and $f_{30}$, it can also be seen that the proposed algorithm is also fast in finding better solutions when solving composition functions.

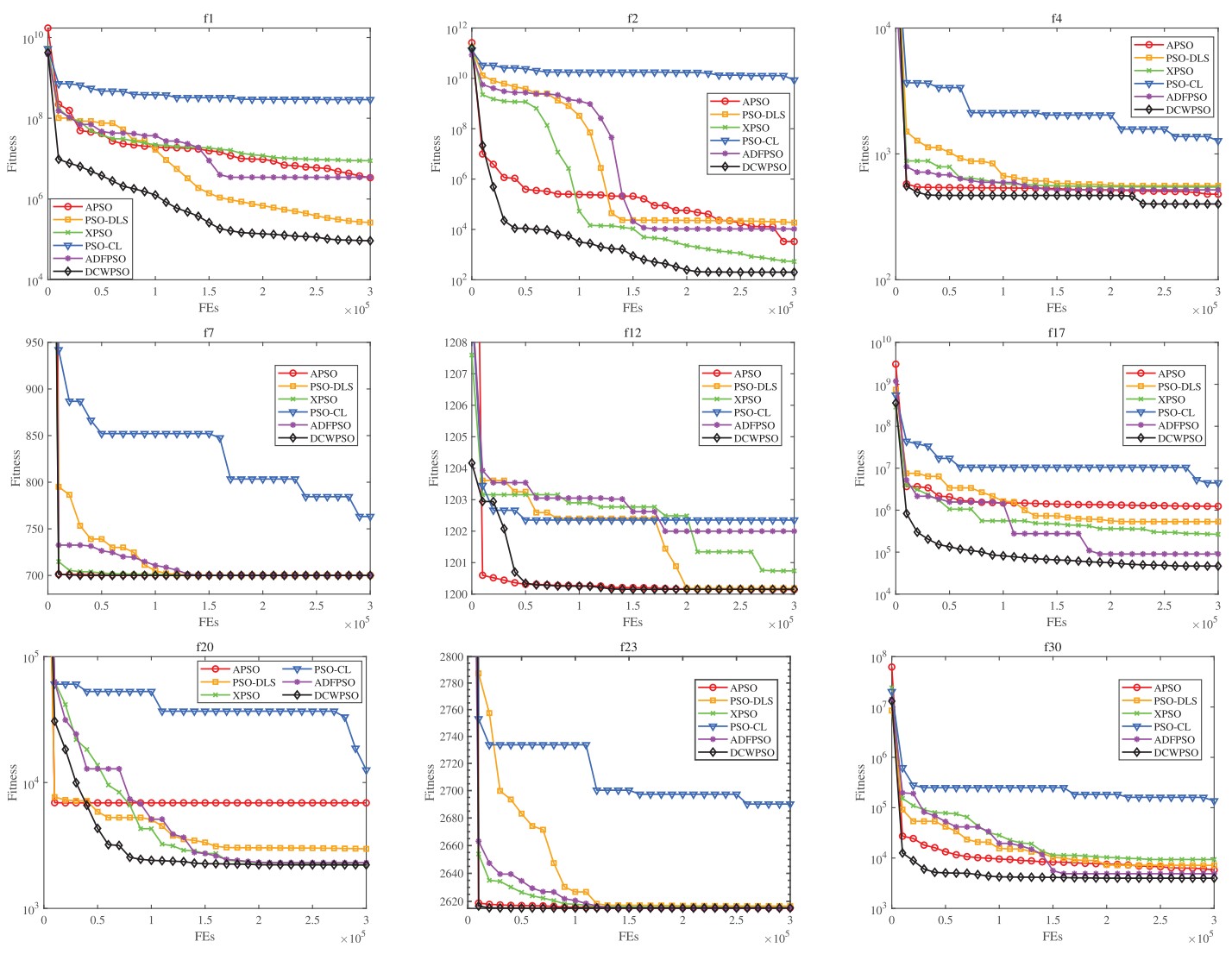

**Figure 4** The convergence curves of DCWPSO compared with PSO variants in the CEC2014 $(30 - D)$.

## Statistical analysis of experimental results

In this section, we employ two widely recognized statistical tests to assess the efficacy of the proposed DCWPSO algorithm compared to other peer algorithms. Specifically, the Wilcoxon sign-rank test (*Derrac et al., 2011*) is utilized to determine whether there exists a significant difference between the performance of DCWPSO and those of other competitors on individual test functions. Additionally, a Friedman test is applied to evaluate the overall performance of all the peer algorithms.

### Wilcoxon sign-rank test

To highlight distinctions between DCWPSO and other PSO variants on the CEC2014 test suites, this study employs the Wilcoxon sign-rank test, a nonparametric statistical analysis

**Table 5 Wilcoxon signed-rank test with different function types on CEC2014 test suite.**

| Function type | Metrics | APSO | PSO-DLS | XPSO | PSO-CL | ADFPSO |
|---|---|---|---|---|---|---|
| Unimodal functions | n/+/−/= | 3/2/1/0 | 3/3/0/0 | 3/3/0/0 | 3/3/0/0 | 3/2/1/0 |
| Simple multimodal functions | n/+/−/= | 13/11/2/0 | 13/6/7/0 | 13/10/3/0 | 13/13/0/0 | 13/12/1/0 |
| Hybrid functions | n/+/−/= | 6/4/2/0 | 6/4/2/0 | 6/6/1/0 | 6/6/0/0 | 6/3/3/0 |
| Composition functions | n/+/−/= | 8/5/3/0 | 8/5/3/0 | 8/5/3/0 | 8/8/0/0 | 8/7/1/0 |
| Overall | /+/−/= | 30/22/8/0 | 30/17/13/0 | 30/23/7/0 | 30/30/0/0 | 30/24/6/0 |

**Table 6 Friedman-test results on CEC2014 test suite.**

| Average rank | Overall | | Unimodal functions | | Simple multimodal functions | | Hybrid functions | | Composition functions | |
|---|---|---|---|---|---|---|---|---|---|---|
| | Algorithm | Ranking | Algorithm | Ranking | Algorithm | Ranking | Algorithm | Ranking | Algorithm | Ranking |
| 1 | ADFPSO | 2.43 | DCWPSO | 1.00 | ADFPSO | 2.23 | DCWPSO | 2.33 | ADFPSO | 2.00 |
| 2 | DCWPSO | 2.67 | XPSO | 2.67 | APSO | 2.62 | ADFPSO | 2.83 | DCWPSO | 3.00 |
| 3 | XPSO | 3.20 | ADFPSO | 3.67 | DCWPSO | 3.00 | XPSO | 3.17 | XPSO | 3.38 |
| 4 | APSO | 3.27 | APSO | 4.00 | XPSO | 3.23 | APSO | 3.50 | APSO | 3.75 |
| 5 | PSO-DLS | 4.40 | PSO-DLS | 4.00 | PSO-DLS | 4.23 | PSO-DLS | 4.17 | PSO-CL | 3.88 |
| 6 | PSO-CL | 5.10 | PSO-CL | 6.00 | PSO-CL | 5.77 | PSO-CL | 5.00 | PSO-DLS | 5.00 |

method. The objective of this test is to scrutinize performance variations between these algorithms.

The results of the Wilcoxon nonparametric test for DCWPSO and these compared algorithms are shown in Table 5. The symbols "n/+/−/=" represents the *n* number of test functions, and that DCWPSO is superior to, inferior to and equals to for comparison, respectively. The table is indexed by various function types in CEC2014. It is evident that the proposed algorithm exhibits only a slight worse in performance in the assessment of simple multimodal functions. Nevertheless, it consistently outperforms other algorithms overall.

### Friedman test

A Friedman test is conducted to provide a comprehensive assessment of the performance of the six algorithms. The results are presented in Table 6, with the algorithms arranged in ascending order based on their ranking values (lower values indicating better performance). Furthermore, separate Friedman tests are performed on the four different types of functions, and the outcomes are detailed in Table 6.

To enhance the presentation of the Friedman test results, we construct a heat map illustrating the performance of all algorithms across four distinct types of test functions and the overall test set in Fig. 5. The visual analysis reveals the outstanding performance of the proposed algorithms in solving unimodal functions and hybrid functions, even though the overall results may not surpass ADFPSO.

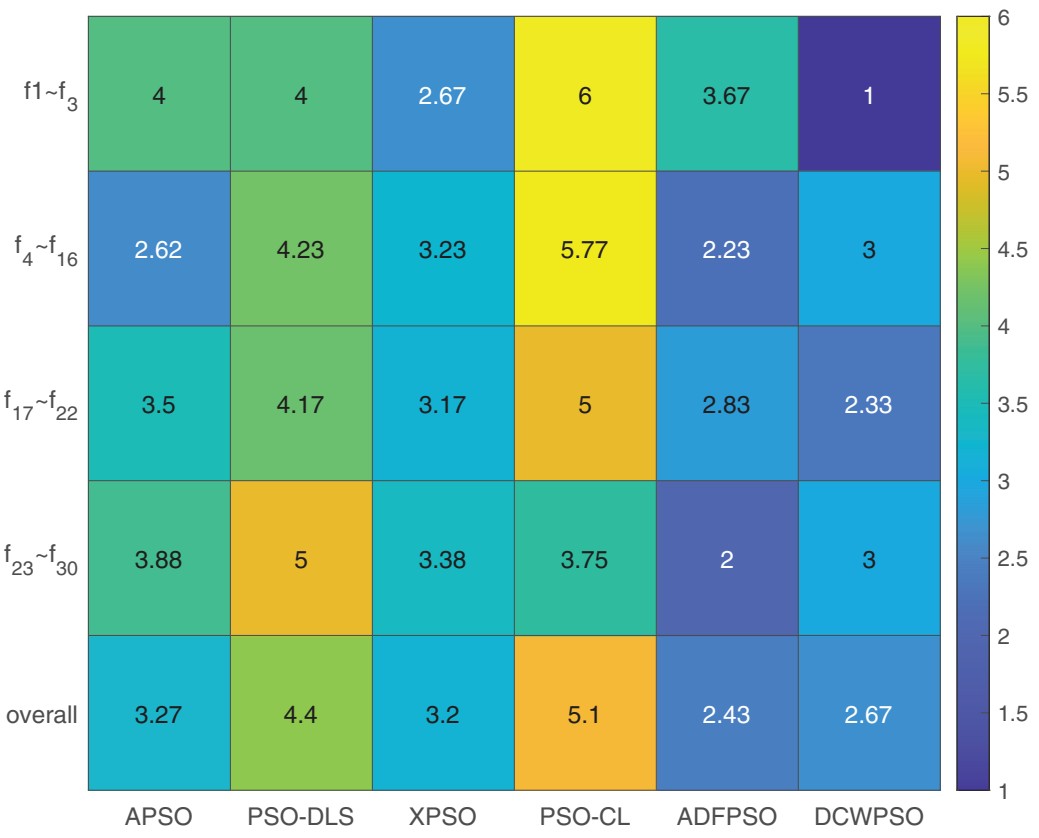

**Figure 5 Heatmap of friedman test results from Table 6.**

## CONCLUSIONS AND FUTURE WORKS

Addressing the drawbacks of conventional PSO, such as premature convergence and susceptibility to local optima, this article formulates a hybrid learning strategy to enhance the performance of the particle swarm algorithm. Firstly, this study proposes a novel dynamic oscillation inertia weight, which produces oscillatory nonlinear inertia weights during iterations. This methodology achieves a more effective equilibrium between algorithmic exploration and exploitation through the modification of the search process. Secondly, this study presents a neighborhood learning strategy and cosine similarity to modify the update of particle velocity based on the observation of cosine similarity between *Pbest* and *Gbest* neighborhoods. Finally, to enhance the overall performance of the entire population, this article introduces the worst-best example learning strategy. This strategy facilitates rapid improvement of the worst *p* particles, contributing to an overall enhancement in effectiveness.

To validate the proposed algorithm's performance, this study conducts experiments comparing performance in accuracy, stability, convergence and statistical analysis. The experimental results indicate that the proposed algorithm generally outperforms peer algorithms in many aspects. However, it is observed that the proposed algorithm shows some limitations in solving multimodal functions. In future work, the integration of

reinforcement learning and deep learning with these learning strategies could better guide the evolutionary direction of particles during the iteration process, thereby enhancing the algorithm's problem-solving capabilities. Furthermore, these algorithms can be applied to address real-world optimization problem.

### Funding

The project is supported by the grant of the Education Department of Hainan Province (Hnky2023-4), the National Natural Science Foundation of China under Grant No. 11861030, and by the Hainan Provincial Natural Science Foundation of China under Grant No. 621RC511. There was no additional external funding received for this study. The funders had no role in study design, data collection and analysis, decision to publish, or preparation of the manuscript.

### Grant Disclosures

The following grant information was disclosed by the authors:
Education Department of Hainan Province (Hnky2023-4).
National Natural Science Foundation of China: 11861030.
Hainan Provincial Natural Science Foundation of China: 621RC511.

### Competing Interests

The authors declare that they have no competing interests.

### Author Contributions

- Yibo Han conceived and designed the experiments, performed the experiments, analyzed the data, performed the computation work, prepared figures and/or tables, authored or reviewed drafts of the article, and approved the final draft.
- Meiting Lin conceived and designed the experiments, performed the experiments, analyzed the data, performed the computation work, prepared figures and/or tables, authored or reviewed drafts of the article, and approved the final draft.
- Ni Li conceived and designed the experiments, performed the experiments, analyzed the data, performed the computation work, prepared figures and/or tables, authored or reviewed drafts of the article, and approved the final draft.
- Qi Qi conceived and designed the experiments, performed the experiments, analyzed the data, performed the computation work, prepared figures and/or tables, authored or reviewed drafts of the article, and approved the final draft.
- Jinqing Li conceived and designed the experiments, performed the experiments, analyzed the data, performed the computation work, prepared figures and/or tables, authored or reviewed drafts of the article, and approved the final draft.
- Qingxin Liu conceived and designed the experiments, performed the experiments, analyzed the data, performed the computation work, prepared figures and/or tables, authored or reviewed drafts of the article, and approved the final draft.

## Data Availability

Code and data are available at GitHub:

https://github.com/P-N-Suganthan/CEC2014

Computer code are also available in the Supplemental Files.

## Supplemental Information

Supplemental information for this article can be found online at http://dx.doi.org/10.7717/peerj-cs.2253#supplemental-information.

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
