# Peer review of "DCWPSO: particle swarm optimization with dynamic inertia weight updating and enhanced learning strategies"

_PeerJ Computer Science, doi:10.7717/peerj-cs.2253_

## Round 0.1 · original submission · Minor Revisions

This is a well written and well researched paper. The reviewers have given only small amounts of feedback and only minor changes to the manuscript are required. When preparing the revised manuscript please pay particular attention to improving the legibility of figures.

Reviewer 3 has given a number of useful suggestions, the authors should carefully consider running the additional test bed functions and adding the requested comparisons even though this will require additional work it will considerably strengthen the resultant manuscript.

Reviewer 1 ·

Basic reporting

In the study, the authors proposed a PSO variant, in which the inertia weight and the learning model are improved.

Experimental design

The experimental results verify the improved PSO can offer promising performance.

Validity of the findings

The new introduced strategies play positive performance on convergence speed and solutions' accuracy.

Additional comments

The performance or the characteristics of new proposed strategies should be analyzed deeply.

Reviewer 2 ·

Basic reporting

- Write an opening paragraph for Section 2.
- Change the title of Section 2 to "Related Work".
- The adverb "where" after an equation should be written with small letters.
- The limitations of the proposed approach should be mentioned in the conclusion section.
- Change the title of the last section to "Conclusions and Future Works".
- The future work directions at the end of the conclusion should be presented in a better way.
- Write an opening paragraph for Section 4.

Experimental design

- The computational complexity of the proposed algorithm should be compared with the original PSO algorithm.
- The proposed algorithm should be compared with additional efficient numerical optimization algorithms: Distributed GWO (DGWO), and island cuckoo search (iCSPM2).

Validity of the findings

- The convergence charts should be discussed with more details.
- The charts in the experimental section should be presented in a better way. Perhaps make them larger and display fewer charts each row.

Reviewer 3 ·

Basic reporting

The authors have proposed the variants of Particle Swarm Optimization to solve various CEC test bed systems.
the paper is nicely written and well organized.

Experimental design

The proposed method should be compared with the latest state of the art WCCI(CEC)/GECCO 2022 test bed functions.
and it should be statistically compared with CEC 2022 algorithms.

Validity of the findings

No comments

Additional comments

No comments

Reviewer 4 ·

Basic reporting

Please improve Figure 3. It is very tiny.

Experimental design

no comment

Validity of the findings

no comment

Additional comments

no comment

---

## Round 0.2 · accepted · Accept

Thank you for carefully revising your manuscript and addressing all of the reviewers concerns.

Reviewer 2 ·

Basic reporting

- No further comments. The paper can be accepted for publication.

Experimental design

- No further comments. The paper can be accepted for publication.

Validity of the findings

- No further comments. The paper can be accepted for publication.